# Post-Transcriptional Control of mRNA Metabolism and Protein Secretion: The Third Level of Regulation within the NF-κB System

**DOI:** 10.3390/biomedicines10092108

**Published:** 2022-08-29

**Authors:** Jasmin Priester, Jan Dreute, Michael Kracht, M. Lienhard Schmitz

**Affiliations:** 1Rudolf Buchheim Institute of Pharmacology, Justus Liebig University, 35392 Giessen, Germany; 2Institute of Biochemistry, Justus Liebig University, 35392 Giessen, Germany; 3Universities of Giessen and Marburg Lung Center (UGMLC), German Center for Lung Research (DZL), 35392 Giessen, Germany

**Keywords:** NF-κB, post-transcriptional regulation, mRNA stability, translation control, ncRNA, protein secretion, tumor microenvironment

## Abstract

**Simple Summary:**

Gene expression regulated by the NF-κB transcription factor pathway is a multi-step process that finally results in the synthesis of proteins, many of which are eventually secreted. The NF-κB system is activated by cytosolic induction pathways (level 1) and then further modulated by cofactors and epigenetic mechanisms in the nucleus (level 2). This review discusses posttranscriptional events as the third level of NF-κB control with a focus on NF-κB-mediated regulation of mRNA metabolism and protein secretion.

**Abstract:**

The NF-κB system is a key transcriptional pathway that regulates innate and adaptive immunity because it triggers the activation and differentiation processes of lymphocytes and myeloid cells during immune responses. In most instances, binding to cytoplasmic inhibitory IκB proteins sequesters NF-κB into an inactive state, while a plethora of external triggers activate three complex signaling cascades that mediate the release and nuclear translocation of the NF-κB DNA-binding subunits. In addition to these cytosolic steps (level 1 of NF-κB regulation), NF-κB activity is also controlled in the nucleus by signaling events, cofactors and the chromatin environment to precisely determine chromatin recruitment and the specificity and timing of target gene transcription (level 2 of NF-κB regulation). Here, we discuss an additional layer of the NF-κB system that manifests in various steps of post-transcriptional gene expression and protein secretion. This less-studied regulatory level allows reduction of (transcriptional) noise and signal integration and endows time-shifted control of the secretion of inflammatory mediators. Detailed knowledge of these steps is important, as dysregulated post-transcriptional NF-κB signaling circuits are likely to foster chronic inflammation and contribute to the formation and maintenance of a tumor-promoting microenvironment.

## 1. The NF-κB System

NF-κB is a collective term for an ancient signaling system that is already existent in organisms basal to insects with functional roles in development and immunity [1]. The entirety of the NF-κB system includes multiple signaling mediators that are important for the principal activation patterns of this pathway and five distinct DNA-binding subunits that form the core of the transcription-activating properties of NF-κB. The NF-κB subunits share an N-terminal NF-κB/Rel homology domain (RHD), which mediates dimerization and DNA binding [2]. Transcriptional induction of target genes depends on the presence of transcriptional activation domains (TADs), which occur in the C-terminal regions of REL (c-Rel), RELB and RELA (p65). Two of the DNA binding subunits, namely NF-κB1 (p105) and NF-κB2 (p100), lack TADs and are produced as precursor proteins [3]. These are processed to yield the DNA-binding forms p50 (derived from p105) and p52 (derived from p100) through phosphorylation-induced partial proteolysis or during translation [4].

The complexity of the NF-κB system is evidenced by the sheer number of signaling components that control release, nuclear translocation and biological activities of the DNA-binding subunits, as well as by their post-translational modifications and hierarchical organization [5,6,7]. Very likely, this complexity has evolved to coordinate the stress response in as many ways as possible [1,8]. Accordingly, there is a wide spectrum of NF-κB activators that range from DNA damage and senescence to pathogen-induced or sterile inflammatory conditions [9]. In the absence of specific induction, NF-κB is found to have only low basal activity, mainly because the subunits are quantitatively retained in the cytosol by inhibitory IκB proteins, which need to be inducibly phosphorylated and proteasomally degraded during the process of NF-κB activation. The multitude of activators is further expanded at the level of at least three different activation pathways, which allow a stimulus-specific and highly tailored response of the NF-κB system. To date, the canonical, non-canonical and atypical NF-κB activation pathways have been identified [5,10].

The canonical pathway is triggered after the recognition of damage-associated molecular patterns (DAMPs) or pathogen-associated molecular patterns (PAMPs) by cytosolic or membrane-anchored receptors [11]. The induced signaling events are transmitted and processed by adapter proteins, including MyD88* (* To facilitate reading of the following sections, abbreviations for proteins are listed at the end.) and TICAM-1/TRIF, several upstream protein kinases such as IRAK1, 4 and ubiquitin E3 ligases, including TRAF6 [12,13]. In response to the initial receptor-ligand interaction, these proteins transiently interact to form multi-protein complexes that ultimately result in the activation of the TAK1 kinase, which in turn activates the so-called IκB kinase (IKK) complex. This complex is composed of two catalytic subunits (IKKα and IKKβ) and the scaffolding protein IKKγ/NEMO [14,15,16]. The IKK complex then phosphorylates the inhibitory IκB proteins, a modification that serves as a recognition mark for the β-TrCP receptor subunit contained in a ubiquitin E3 ligase complex which mediates subsequent IκBα K48-ubiquitination and enables its proteasomal degradation [17].

The non-canonical NF-κB activation pathway is of special relevance for B cell survival and lymphoid organ development. A central step in non-canonical NF-κB signaling is the stabilization of the protein kinase NIK and the activation of its downstream kinase IKKα, which leads to C-terminal processing of the NF-κB2/p100 precursor that also functions as a cytoplasmic IκB protein. Activation of the non-canonical pathway leads to the formation of p52/RelB dimers, a process that occurs with much slower kinetics as compared to activation of the canonical pathway [18,19].

The atypical NF-κB pathway is induced by DNA damage, as it occurs after chemo- and radiotherapies or the expression of oncogenes. Mild DNA damage leads to the induction of proliferative arrest and senescence after several days. During the rapidly activated first phase, DNA damage is sensed by PARP, which, together with the DNA damage responsive kinase ATM, triggers a signaling cascade that leads to activation of TRAF6 and monoubiquitination of NEMO [20,21]. These events route the signal into the IKK pathway and mediate proteasomal destruction of the inhibitory IκBα protein, thus promoting the expression of anti-apoptotic genes. The first phase is terminated by a regulatory feed-back loop that leads to IκBα re-expression from the *NFKBIA* gene. Prolonged DNA damage allows the induction of a second phase that is IKK-independent and attributable to repression of the *NFKBIA* gene and strongly reduced IκBα levels [22].

## 2. The Concept of the Third Level of NF-κB Regulation

Full execution of the NF-κB pathway requires different layers of regulation. As discussed above, the various cytosolic activation pathways have in common that they result in the formation and release of dimeric DNA-binding subunits. In addition to this cytosolic level, NF-κB regulation also occurs in the nucleus, where various molecular mechanisms shape the NF-κB response [23]. These include the control of (I) chromatin accessibility to expose NF-κB motifs, (II) regulated genome-wide 3D interactions of NF-κB-bound enhancers and promoters, and (III) the co-recruitment of additional transcription factors, mediators and cofactors to chromatin-associated NF-κB [24,25]. *Vice versa*, NF-κB also contributes to the regulation of chromatin modifications and compaction, for example by recruiting cofactors such as AKIRIN2 and IκBζ, which in turn connect to the chromatin remodeling complex SWI/SNF [26]. NF-κB subunits are also crucially important for chromatin recruitment of CBP/p300, which are the principal H3K27 acetyl transferases, a histone mark that characterizes highly active enhancers [25,27,28,29,30,31]. Thus, there is a wealth of information on the chromatin-associated processes that are regulated by NF-κB. Additionally, there is ample evidence that post-translational mechanisms (phosphorylation, ubiquitination, acetylation) regulate NF-κB activation and function [32,33,34,35].

In comparison, there is less knowledge on how NF-κB modulates the steps of post-transcriptional gene regulation and protein secretion. In this review, we discuss emerging evidence for this third layer of NF-κB regulation at the level of post-transcriptional gene regulation, which encompasses all steps of the mRNA life cycle, including protein maturation and secretion.

We suggest that activities of NF-κB operating at multiple levels beyond transcription offer a number of key advantages to achieving an optimized immunoregulatory gene response. At present, the following scenarios illustrate some of the benefits of additional post-transcriptional control within the NF-κB system: (I) An effective immune response is initially triggered by recognition of pathogens or damage, followed by the effector and the termination phase. The latter is characterized by the resolution of inflammation governed by both rapid downregulation of pro-inflammatory factors (such as IL-1β, IL-6, TNFα, IFNγ, CXCL8) as well as increased expression and secretion of anti-inflammatory factors (such as IL-10). This is followed by the final phase of tissue repair and remodeling [36,37,38]. Due to these sequentially ordered phases, the mechanisms discussed in this review are likely to play an important physiological role in the sophisticated temporally staggered expression of inflammatory mediators that coordinate the overall inflammatory process [39]. (II) NF-κB-dependent regulation of post-transcriptional events will allow rapid adjustment of mRNA copy numbers in the cytoplasm by modulating their stability (or accelerated decay) via the same pathway. In this sense, post-transcriptional regulation may accelerate the silencing of the prototypical NF-κB-driven gene response by efficiently degrading mRNA copies of NF-κB targets after their transcription is terminated [40]. (III) Isolated increases of mRNA stabilities of NF-κB target genes in the absence of altered transcription rates can theoretically provide a mechanism for upregulation of inflammatory mediators that is independent of the nuclear NF-κB pathway, allowing to bypass the energy-demanding and tedious process of transcription. Additionally, the spectrum of NF-κB mRNA targets may be expanded beyond genes solely controlled by NF-κB subunits. For example, some upstream components of the NF-κB system (TRAF6, IKKs) have been shown to bypass the nucleus and regulate components of mRNA decay pathways directly. (IV) Spurious transcription bears the risk of inadequate expression of inflammatory mediators that may cause smoldering inflammation in chronic inflammatory diseases or in the tumor microenvironment (TME). Post-transcriptional control by the NF-κB system offers a mode of reducing this transcriptional “noise” [41]. (V) By simultaneously affecting both mRNA synthesis and mRNA decay rates, multi-level control by NF-κB ensures transcript-specific “buffering”. Buffering of mRNA levels is an emerging mechanism that helps to scale gene expression according to the metabolic capacities of cells and thus confers increased robustness of the system [42,43]. (VI) Finally, NF-κB-dependent control of translation and protein secretion will allow the entire process of gene expression to be coordinated in its final step to achieve the biologically desired production of immunoregulatory factors at the right time and at the required extracellular concentrations [44,45].

In summary, we suggest that post-transcriptional regulation is an integrated feature of the NF-κB response rather than mediated by independent molecular machines and cellular processes, as schematically summarized in Figure 1. In the following text, we discuss examples of key molecules that have been suggested to participate in the various mechanisms described above.

## 3. The Role of the NF-κB System in Posttranscriptional Gene Regulation

### 3.1. Key Aspects of the Post-Transcriptional Regulation Relevant to Regulation of NF-κB Target Genes

Nascent mRNAs are co-transcriptionally capped at the 5′ ends, while the 3′ ends are cleaved and polyadenylated. These transcripts are spliced to remove the introns and already associate co-transcriptionally with many further proteins to form ribonucleoprotein particles (mRNPs), followed by export from the nucleus [46,47]. All of these early steps are highly regulated in response to signaling events [48]. In the cytoplasm, mRNAs can be stored, undergo regulated decay or become translated to proteins at the ribosome [49]. Finally, a significant proportion of the synthesized proteins reaches the extracellular space by various secretory pathways. All of these steps are strongly regulated in the context of inflammatory gene expression [32,50,51,52].

The stability of many immunoregulatory mRNAs is controlled by proteins binding to specific sequences such as adenine and uridine-rich elements (AREs) that are typically more frequent in the 3′ untranslated regions (UTRs) [53]. A systematic analysis of mRNAs coding for mediators of inflammation showed an overrepresentation of AREs in inflammatory transcripts and an inverse correlation between the number of AREs and transcript stability [45]. The AREs are bound by different RNA-binding proteins, including TTP and HuR [54]. The binding of TTP stimulates a process called ARE-mediated mRNA decay (AMD), either upon delivery to a complex of 3′-5′ exonucleases termed the exosome or by 5′-3′-mediated mRNA decay through the 5′-exonuclease XRN1 [55]. These two pathways mediate the bulk of mRNA decay, but additionally, endonucleolytic mRNA decay is also used to degrade transcripts [56].

The components of the 5′-3′-mRNA decay pathway are prevalent in the cytosol but also aggregate in membrane-less organelles called processing bodies (P-bodies), microscopically visible macro-molecular complexes that contain multiple other RNA binding proteins, cellular factors of various functions and non-translating mRNAs [57,58,59]. For a long time, P-bodies were thought to mediate specific forms of mRNA decay, but the current picture is that they are more relevant for transient mRNA storage of translationally repressed mRNA during different forms of stress [60,61,62]. Later on, when the stress has ceased and cells resume function, specific mRNAs contained in P-bodies can be released to become degraded in the cytosol or to enter translation [63,64,65].

The 5′-3′ mRNA decay pathway is well characterized [66,67]. After removal of the 7-methyl guanosine-diphosphate cap structure by the decapping enzyme DCP2 and its regulatory subunit DCP1a, the 5′-3′ exonuclease XRN1 can degrade the mRNA to individual nucleotides starting from the exposed mono-phosphorylated 5′ end [68,69,70]. In human cells, the decapping activity also requires EDC4, which is an essential scaffolding protein for the assembly of P-bodies [69,71]. Cell stress, including inflammation, leads to an increase in the number of P-bodies [72,73,74]. P-bodies are related to stress granules, but a distinguishing criterion is the presence of components of the translation initiation machinery, such as eukaryotic initiation factor 3 (eIF3) subunits in stress granules [59,75,76]. As the name implies, the formation of stress granules occurs in response to cellular stress, such as translation inhibition, and these assemblies selectively concentrate translationally silenced mRNAs [77]. While binding of TTP to AREs typically results in mRNA decay [78], association with HuR does not lead to mRNA destabilization, probably through competition for binding with factors such as TTP [79,80]. In addition, HuR can either inhibit or promote the translation of ARE-containing mRNA molecules [81,82]. Despite these transcript-specific effects, animal models showed that HuR functions as a negative modulator of inflammation by diminishing the translation of cytokine mRNAs in synergy with the translational silencer TIA-1 [83]. Tethering of regulatory proteins to mRNAs can also occur in an ARE sequence-independent manner, e.g., by recognition of stem-loop structures. An example of such a protein is the endoribonuclease Regnase-1, which is also called ZC3H12A or MCPIP1 [84]. Regnase-1 interacts with stem-loop structures present in the 3′ UTR of cytokine mRNAs such as *IL6*. This protein also harbors a domain conferring RNase activity that is triggered upon translation termination to result in mRNA cleavage. As a consequence, Regnase-1 functions to dampen the expression of inflammatory mRNAs. As Regnase-1 is rapidly degraded after stimulation of cells with TLR ligands or IL-1β, proteolytic elimination of this negative regulator allows for unrestricted expression of inflammatory mRNAs [85].

Another important axis of post-transcriptional gene regulation is mediated by non-coding RNAs (ncRNAs), microRNAs (miRNAs) and long ncRNAs (lncRNAs). miRNAs bind to target mRNAs with partial complementarity, and thus, one miRNA allows interference with many different targets. The mRNA-bound miRNAs cause decay as well as inefficient translation of the target transcript [86]. The ever-growing number of miRNA targets with a role in NF-κB signaling and inflammatory processes, including miR-146a, are presented in a number of comprehensive reviews [87,88]. Compared to miRNAs, thousands of lncRNAs encoded in our genome have a broader range of capabilities to interfere with different steps of gene expression. In this context, lncRNAs can influence all steps of gene expression (transcription, splicing, and translation) by engaging in RNA/RNA and RNA/protein interactions. LncRNAs affecting inflammation include lincRNA-Cox2, HOXA-AS3 and lincRNA THRIL, which act by a multitude of different mechanisms [89].

### 3.2. NF-κB-Mediated Regulation of Post-Transcriptional RNA Processing and Protein Translation

The NF-κB system regulates post-transcriptional gene expression either by transcription-dependent or -independent mechanisms. Transcriptional NF-κB targets comprise miRNAs including miRNA146, miRNA155, miR-18a-3p, miR-4286 and miR-19a-3p [90,91,92], as well as lncRNAs such as MALAT1, LINC00665 and PINT [93]. Interestingly, it was recently shown that miR146a and miR155 are antagonistic parts of a miRNA-based regulatory network that fine-tunes NF-κB activity in macrophages. While increased miR155 expression potentiates NF-κB activity in miR-146a-deficient mice, enforced miR-155 expression counteracts miR-146a-mediated repression of NF-κB activation. Overall, this study revealed that miRNA-based regulatory networks participate in the kinetic control of inflammatory responses [94].

These few examples emphasize the potential connections between NF-κB and miRNAs or lncRNAs, but this field is complicated by the multiplicity of targets and the mutual interactions between different classes of ncRNAs. Most likely, the NF-κB-controlled ncRNAs serve to remodel the output program of inflammatory signaling. Since the distinct functions of NF-κB-regulated ncRNAs have not been comprehensively characterized, targeted deletion or (over)expression of these ncRNAs in intact organisms or cellular models will be required for a deeper understanding of their function and mutual interaction.

NF-κB-mediated regulation by non-transcriptional mechanisms may be faster than transcription-dependent processes and also allows the expansion of the spectrum of regulated factors. Transcription-independent regulation by the NF-κB system has been described for several mRNA-bound proteins, namely TTP, HuR and Regnase-1, as visualized in Figure 2. TTP can be phosphorylated by various kinases, including the NF-κB activating kinase MEKK1 at N-terminal residues [95]. These TTP phosphorylations are the prerequisite for subsequent TRAF2-mediated K63-linked ubiquitination at the zinc finger of this RNA-binding protein. This regulatory ubiquitination provides a functional switch to enable prolonged activation of JNK in response to TNF stimulation [96]. Additionally, the HuR protein can be modified at multiple sites, including IKKα-mediated phosphorylation at S304. This modification creates a docking site for the E3 ubiquitin ligase adapter β-TrCP1 and enables subsequent signal-regulated ubiquitin/proteasome-mediated degradation of this RNA-binding protein [97]. Similarly, also the expression levels of Regnase-1 can be controlled by a phosphorylation-dependent ubiquitination event. Regnase-1 inducibly associates with IKKβ following activation of TLRs or the IL-1 receptor, allowing for inducible phosphorylation of this RNA stem-loop binding protein. This, in turn, enables docking of β-TrCP and ubiquitin/proteasome-dependent elimination of this RNA-cleaving enzyme [85]. This degradation allows indirect IKK-mediated control of the stability of Regnase-1 associated mRNAs, including those coding for *IL6* and *IL12b*. Regnase-1 is also phosphorylated by its interactor IRAK1, and this modification facilitates inducible ubiquitination of Regnase-1 by an uncharacterized mechanism [85].

In addition to RBPs, P-bodies are subject to NF-κB-mediated regulation, as summarized in Figure 3. NF-κB activation enhances the interaction of IKKγ with EDC4 and allows IKK-mediated phosphorylation of EDC4 on several serines in the WD40 domain and in the serine-rich linker. The interaction between the IKK complex and P-bodies was also seen at the cellular level, as detected by increased co-localization between P-bodies and IKKβ or IKKγ as well as by a higher number of P-bodies after NF-κB activation [98]. IKK-mediated EDC4 phosphorylation was required for inducible co-recruitment of the P-body components DCP1a and DCP2 and also for the stimulus-dependent increase in P-body numbers. In addition, different experimental approaches revealed that IKKβ and also its kinase activity are required for the increase in P-body numbers. Deletion of EDC4 and/or IKKβ also affected the stability of many transcripts, approximately a third of which, including the *IL8* mRNA, showed regulation by both factors. While EDC4 destabilized the *IL8* transcript in the absence of stimulation, IKK-mediated phosphorylation of EDC4 resulted in an increased half-life of this mRNA, adding a further mechanism for the control of steady-state levels of mRNAs encoding *IL8* and further factors [98]. Interestingly, the majority of transcripts whose stability was regulated by IKK do not derive from annotated NF-κB target genes, demonstrating a significant expansion of the IKK-regulated mRNA spectrum. EDC4 also shows inducible interaction with TRAF6 [98], but this association might be indirect, as this E3 ligase is known to interact with DCP1a to mediate modification of this decapping enzyme with K63-linked ubiquitin chains at multiple sites. DCP1a ubiquitination is essential for the formation of P-bodies and the interaction with DCP2, EDC4, and XRN1 [99]. This type of ubiquitin chain branching is not only required for the remodeling of P-body architecture but also for the decay of specific mRNAs, including *IL8*, as revealed by the analysis of genetically engineered cells incapable of forming K63-linked chains [99]. In the same study, TRAF6 was also shown to indirectly regulate DCP1a phosphorylation, a modification relevant for the formation of P-bodies in response to cytokines or stressors [74]. As many interactions of protein kinases with their substrates are inducible and transient [100], we anticipate that recent advances in proximity interactomics [101] will extend the number of NF-κB interactors with a function in the mRNA decay pathways and mRNP assemblies.

It is well established that inflammatory gene expression is also regulated at the level of ribosomal translation [102,103], but the information on NF-κB-derived signals for *de novo* protein synthesis is scattered and so far limited (Figure 4). One well-studied example for NF-κB-mediated control of translation is provided by the finding that lipopolysaccharide (LPS)-induced phosphorylation of eIF4E is largely diminished in macrophages of mice lacking the activity of IRAK2- or IRAK4 [104,105]. Accordingly, the number of LPS-induced mRNAs associated with actively translating ribosomes was reduced in IRAK2-deficient macrophages [105]. NF-κB can also affect translation elongation by its ability to repress the expression of eEF2K. Reduced levels of this kinase consequently resulted in diminished T56 phosphorylation of eEF2, which alleviated the repressive function of this modification on translation elongation [106]. At present, a comprehensive understanding of the global impact of NF-κB-derived signaling on *de novo* protein synthesis is lacking, but we expect that recent advances in quantitative translation proteomics will allow us to fill this gap [107,108,109].

## 4. The Role of the NF-κB System in Protein Secretion

### 4.1. Key Aspects of Conventional and Nonconventional Protein Secretion

A substantial fraction of NF-κB target genes is translated to proteins that reach the extracellular space by conventional or unconventional secretion. The secreted factors mediate physiological processes to maintain homeostasis but are also relevant in pathophysiological situations such as infection and inflammation or in tumors where the tumor secretome creates a pro-inflammatory TME [110,111]. In these scenarios, mediators of inflammatory processes, including chemokines and cytokines, are released into the extracellular space in order to trigger autocrine and paracrine signaling [112]. The process of protein secretion is tightly regulated in order to avoid excessive release of inflammatory mediators, as they occur, for example, by mutations triggering constitutive inflammasome activity [113]. Proteins released by conventional secretion harbor a leader peptide that enables binding to the signal recognition particle (SRP), followed by protein synthesis at ER-bound ribosomes [114]. The cargoes exit the ER via COPII-coated vesicles to reach the Golgi apparatus, where the proteins can be modified further. Exit at the trans-Golgi and transport to the cell membrane occurs in COPI-coated vesicles [115].

In addition to conventional secretion, cargoes can be released by a number of different routes that are collectively referred to as unconventional secretion [116,117,118]. These unconventional secretion pathways have been systematically categorized, as visualized in Figure 5, and can be typically induced by stress, including inflammatory events [51]. Mass spectrometric analysis of the secretome from TLR4-induced macrophages showed that approximately one-third of the 775 inducibly released proteins lack a signal peptide, suggesting a broad relevance of unconventional secretion [119]. An unconventional secretion pathway for a leader peptide-containing cargo has been found for membrane proteins or secreted factors that can be exported on different routes that bypass the Golgi complex on their way to the cell surface (Type IV secretion). All other non-conventional secretion modes can mediate the export of leaderless cargoes, which are synthesized at cytosolic ribosomes. Type I secretion occurs by direct translocation of cytosolic protein across pores in the plasma membrane, as exemplified by the release of FGF2 [118,119]. Type II secretion is mediated by plasma-membrane-resident ABC transporters but is not discussed further, as this process is not explored in humans [117]. Type III secretion proceeds by uptake of exported proteins, including IL-18 and LIF, into endocytic-membrane-enveloped compartments such as lysosomes, autophagosomes, endosomes and exosomes [120]. Recent work discovered a further pathway of unconventional protein secretion emanating at the ER-Golgi intermediate compartment (ERGIC). Here, leaderless proteins are translocated into the ERGIC vesicles by TMED10 protein channels, allowing vesicle-mediated export that bypasses the Golgi compartment [121].

### 4.2. NF-κB-Mediated Regulation of Protein Secretion

The contribution of NF-κB to secretory pathways is schematically displayed in Figure 6, but our understanding is rather limited, which can be attributed to several confounding factors: (I) Cells typically employ various secretory pathways in parallel. (II) Mass spectrometric identification of the secretome is technically challenging, and in addition, it is difficult to distinguish between a regulated secretion event and the unspecific release by cell damage. (III) In many reports claiming a contribution of NF-κB to the secretion of a given protein, the (well-known) transcriptional effects were not distinguished from the subsequent levels of regulation. Studies determining both processes in parallel revealed that the direction and magnitude of transcription and secretion show no linear correlation for the majority of targets [119,122]. (IV) Secretion of many proteins occurs in a cell-type-specific fashion. While each nucleated cell has the principal capacity to secrete proteins, specialized cell types such as eosinophils, mast cells or granulocytes have developed highly cell-type-specific exocytotic mechanisms [33]. (V) A particular cytokine can be secreted by different pathways, as impressively illustrated by the release mechanisms that have been described for IL-1β. This cytokine is produced as an inactive precursor that can be cleaved by caspase-1 contained in the NLRP3 inflammasome or by caspase-1-independent mechanisms [123,124] to yield the mature IL-1β protein. Release of IL-1β via the type I pathway employs pore formation by the N-terminal fragment of gasdermin D to allow plasma membrane passage [125,126]. In addition, IL-1β can be released via the type III pathway by release from vesicular carriers, including multivesicular bodies, autophagosomes and secretory lysosomes [127,128,129] and also by the TMED10 pathway [121].

The signaling networks leading to the employment and upregulation of (unconventional) protein secretion are just emerging and involve changes in the lipid composition of vesicles, the redox status and pathways triggered by integrins and cytokines [130,131]. While we currently lack evidence for a transcription-independent role of NF-κB on protein secretion, a negative role of NF-κB encoded genes for the regulation of IL-1β secretion has been revealed in mouse models. In LPS-stimulated myeloid cells, deletion or pharmacological inhibition of IKKβ resulted in elevated plasma levels of IL-1β due to increased processing of pre-IL-1β [132]. In myeloid cells, NF-κB directs the expression of genes such as PAI-2 and CARD18 (also called ICEBERG), whose products limit caspase-1 activation by uncharacterized mechanisms [132]. Additionally, the anti-apoptotic NF-κB target gene Bcl-xL binds and inhibits NLRP1, thus suppressing activation of caspase-1 and processing of pre-IL-1β [133]. The situation is different in neutrophils, where LPS-induced secretion of IL-1β is caspase-1-independent and rather mediated by Proteinase 3 (PR3) and neutrophil elastase (NE) [123]. These proteases can be negatively regulated by NF-κB target genes via unknown mechanisms that might employ the upregulation of several serine protease inhibitors (Serpins) [132]. It is currently incompletely understood whether NF-κB-encoded genes also regulate the processing and transport of further cytokines in addition to IL-1β.

A further candidate protein from the NF-κB pathway with a role in the regulation of secretion is the ubiquitin E3 ligase ITCH, which limits NF-κB activity by several mechanisms, including the degradation of TAK1 [134] and controlling the function of the ubiquitin-editing enzyme A20 [135]. A novel function of ITCH for the process of IL-8 and IL-6 secretion was identified in a siRNA screen performed in oncogene-transformed lung fibroblasts that constitutively produce a pro-tumorigenic secretome [122]. It will therefore be exciting to learn more about the molecular mechanisms employed by ITCH in the future.

## 5. Conclusions

In this review, we have discussed examples demonstrating that the long arm of NF-κB extends far (and beyond transcription) to control all known levels of gene expression. We propose a framework of three levels of regulation intertwined within the NF-κB system. This concept provides a refined explanation for the temporally and quantitatively coordinated production of intracellular or extracellular proteins in immune responses that are controlled by further feed-forward and feed-back loops. These aspects are of high clinical relevance, as inadequately controlled constitutive secretion of inflammatory mediators drives chronic sterile inflammation in diseases such as arthritis. In addition, dysregulated expression and secretion of inflammatory mediators is important in various cancers, where the tumor and stromal cells interact to establish a pro-tumorigenic TME. Since the tumor secretome has emerged as a key factor in tumorigenesis and therapy resistance in this context, identification of the involved signaling pathways and components will pave the way for the development of specific inhibitors that might be of therapeutic use.

## Figures and Tables

**Figure 1 biomedicines-10-02108-f001:**
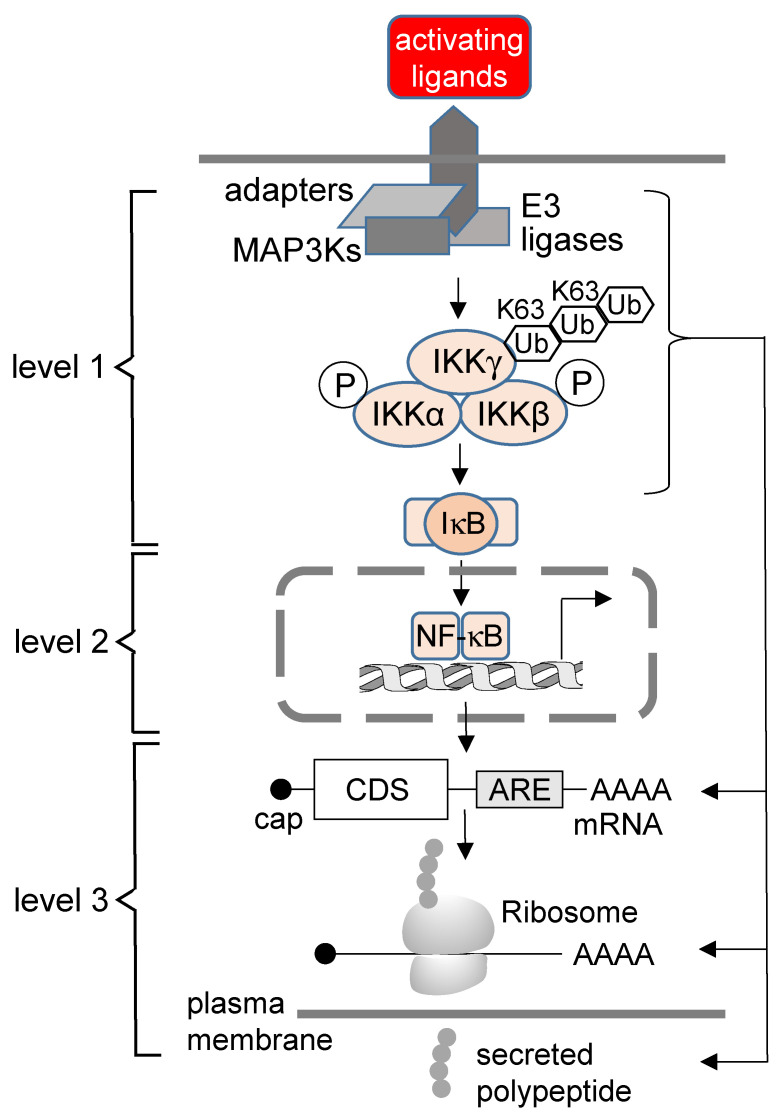
The concept of three levels of regulation in the NF-κB system. For details, see text (CDS, coding sequence; MAP3K, mitogen-activated protein kinase kinase kinase).

**Figure 2 biomedicines-10-02108-f002:**
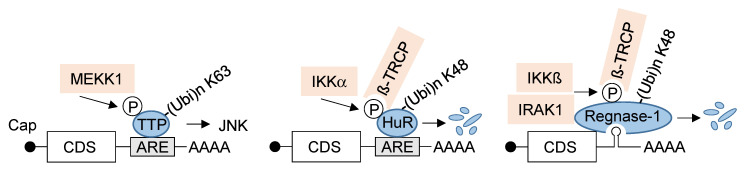
Regulation of mRNAs by RNA-binding proteins. The RNA-binding proteins are depicted in blue, and regulators of the NF-κB pathway are shown in light red; the consequences of the signal-regulated modifications of RBPs are indicated.

**Figure 3 biomedicines-10-02108-f003:**
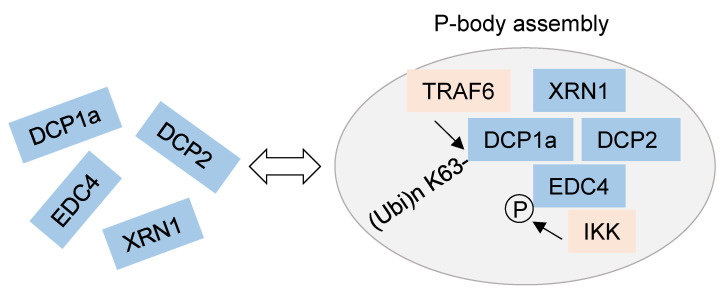
The impact of NF-κB-derived phosphorylations and ubiquitinations on P-body assembly and function. Regulators of the NF-κB pathway are shown in light red; for further details, see text.

**Figure 4 biomedicines-10-02108-f004:**
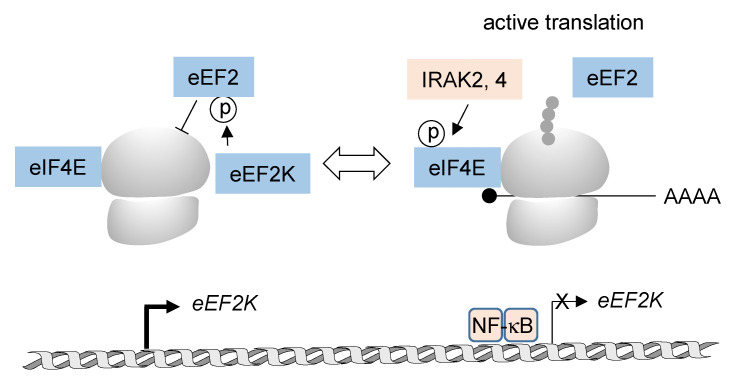
The impact of NF-κB-mediated gene expression and IRAK2/IRAK4 function on protein translation, the ribosomes and the nascent peptide chain are schematically depicted.

**Figure 5 biomedicines-10-02108-f005:**
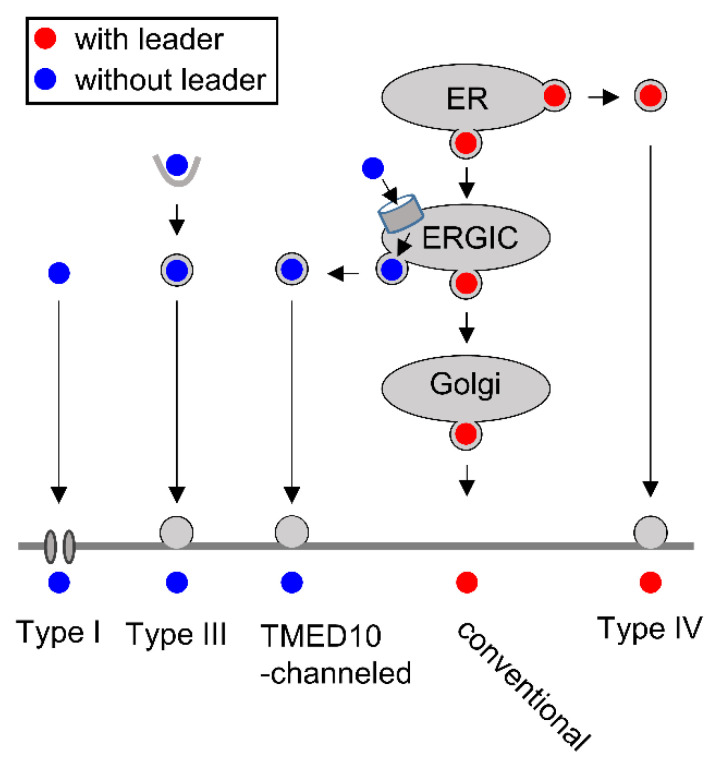
Pathways of protein secretion. Proteins with or without a leader are indicated. The type II secretion pathway is not displayed, as its occurrence and relevance in humans are not known, and also, the formation of lipid-bound extracellular vesicles is not visualized, as this process constitutes a separate research field.

**Figure 6 biomedicines-10-02108-f006:**
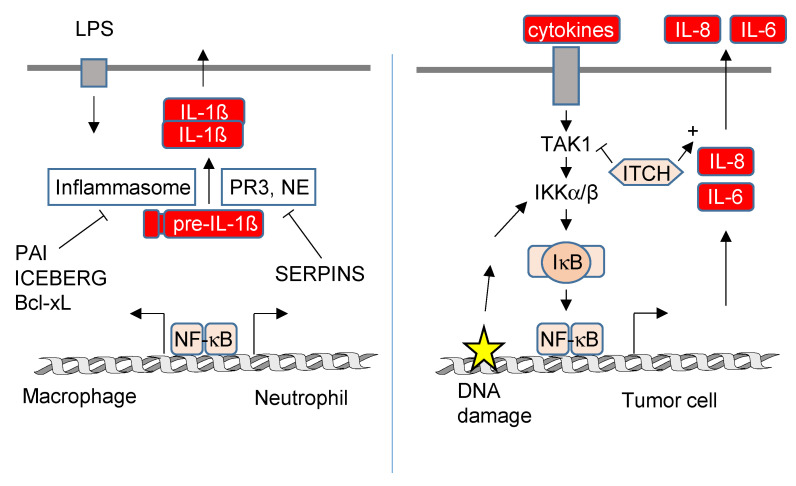
Regulation of protein secretion by NF-κB. The left part visualizes the negative function of NF-κB target proteins on pre-IL-1ß processing proteases and IL-1ß release in macrophages and neutrophils. The right part shows the contribution of ITCH to the NF-κB pathways in senescent tumor cells that activate NF-κB by autocrine mechanisms and by ongoing DNA damage. ITCH also supports secretion of *IL6* and *IL8* by unknown mechanisms.

## Data Availability

Not applicable.

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
