# Peer review of "Post-Transcriptional Control of mRNA Metabolism and Protein Secretion: The Third Level of Regulation within the NF-κB System"

_biomedicines, 2022, doi:10.3390/biomedicines10092108_

Round 1

Reviewer 1 Report

In the manuscript by Knauff et al., the authors review how NF-kB (-related) proteins regulate the expression and secretion of inflammatory mediators on a posttranscriptional level. The review is well-structured and discusses the regulation of mRNA stability/accessibility, translational control, and protein secretion by NF-kB members. The manuscript is easy to comprehend and includes the most relevant citations to the topic. Even though there are numerous reviews covering the molecular actions of NF-kB, the special focus on the posttranscriptional function of NF-kB is interesting and rather novel. Except for a minor comment, I can recommend the review for publishing.

+ Abbreviations used in the figures should be explained in the figure legends (e.g. figure 6, “NE”)

+ page 12: “… and Caspase recruitment domain-containing protein 18 CARD18”; please add brackets

Author Response

1) Abbreviations used in the figures should be explained in the figure legends (e.g. figure 6, “NE”)

Answer: Following this suggestion we give abbreviations for proteins shown in the figures where appropriate, we created a new section 6 which lists all abbreviations for proteins.

2) page 12: “… and Caspase recruitment domain-containing protein 18 CARD18”; please add brackets

Answer: This mistake was corrected.

Reviewer 2 Report

The article entitled “Post-transcriptional control of mRNA metabolism and protein secretion: The third level of regulation within the NF-κB” presents some novel insights into the NF-κB system on the level of posttranscriptional gene expression and secretion of proteins involved in the inflammation process.

The article is well written. The Introduction is informative, concise, and presents crucial background information. Sections are described in detail, clearly, and satisfyingly. The article is gripping, comprehensive yet not overloaded with details, and pleasant to read.

Overall, I recommend it for publication after only some minor improvements.

Below, minor remarks to improve the manuscript are listed point by point.

·       Section 1 would much benefit if a schematic Figure would be added to illustrate the system

·       Please use lower case letters to indicate numbering – in sections 1, 3, and 6 upper case letters were used while in section 2 lower case letters were used and it seems that number (ii) is missing in section 2.

·       Please explain all abbreviations used in figures in the figure captions

·       Figures would benefit if all elements were marked e.g., DNA, RNA, etc.

Author Response

1) Section 1 would much benefit if a schematic Figure would be added to illustrate the system

Answer: We use the amended Figure 1 to explain the NF-kB system and its regulation at various levels.

2) Please use lower case letters to indicate numbering – in sections 1, 3, and 6 upper case letters were used while in section 2 lower case letters were used and it seems that number (ii) is missing in section 2.

Answer: This mistake is now corrected.

3) Please explain all abbreviations used in figures in the figure captions.

This was changed as requested.

4) Figures would benefit if all elements were marked e.g., DNA, RNA, etc.

We marked these items, see also the changes in Fig. 1.

As part of the revision we have made other changes and improvements, all of which are visible in tracking mode. In addition, there were changes in the weighting of the relative contribution of authors, these are consented and agreed among the authors. Our first author Jasmin Knauff has married and thus we give her new last name as Jasmin Priester.

Looking forward to your reply, with best wishes,

Lienhard Schmitz